# Gluteus medius muscle activation patterns during gait with Cerebral Palsy (CP): A hierarchical clustering analysis

**Mehrdad Davoudi, Firooz Salami, Robert Reisig, Katharina S. Gather, Sebastian I. Wolf** *

Clinic for Orthopaedics, Heidelberg University Hospital, Heidelberg, Germany

* Sebastian.Wolf@med.uni-heidelberg.de

**Data Availability Statement:** All relevant data are within the manuscript and its Supporting Information files.

**Funding:** German Research Foundation (DFG) no: WO 1624/ 8-1.

## Abstract

Duchenne gait, characterized by an ipsilateral trunk lean towards the affected stance limb, compensates for weak hip abductor muscles, notably the gluteus medius (GM). This study aims to investigate how electromyographic (EMG) cluster analysis of GM contributes to a better understanding of Duchenne gait in patients with cerebral palsy (CP). We analyzed retrospective gait data from 845 patients with CP and 65 typically developed individuals. EMG activity of GM in envelope format were collected and examined with gait kinematics and kinetics parameters in frontal plane and hip abductor strength, and hip abduction passive range of motion. Six key EMG envelope features during ten gait phases were extracted and normalized. A hybrid K-means-PSO clustering algorithm was employed, followed by hierarchical clustering. The identified clusters were characterized by having a low (cluster_1), medium (cluster_2), and high (cluster_3) activity of GM during loading response. The patients in cluster_1 also exhibited pathological gait characteristics, including increased trunk lateral lean and weak hip abductor, which are associated with Duchenne gait. The patients in this cluster were subclustered according to their response to the intervention: SUB_1 with a significant improvement in trunk obliquity, pelvic obliquity, and hip abduction after intervention, and SUB_2 without such improvement. Comparing pre-treatment EMG and clinical exam of the sub_clusters, SUB_1 had significantly higher activity of GM during 50–87% of the gait cycle with a greater passive range of hip abduction compared to SUB_2. This study established a relationship between EMG of GM and frontal plane gait abnormalities in patients with CP, highlighting potential improvement in Duchenne gait with prolonged GM activity during swing after the intervention.

## 1. Introduction

Hip abductor muscle dysfunction is frequently observed in pediatric cerebral palsy (CP), contributing to an altered trunk, pelvic and hip movement, including an excessive medio-lateral trunk lean [1]. It leads to increased energy demands during walking, which can reduce endurance and limit participation in daily activities [2]. Furthermore, it may induce functional

**Competing interests:** The authors have declared that no competing interests exist.

restrictions, hindering age-appropriate tasks and basic activities of daily living such as walking, dressing, and playing [3]. Weakness or spasticity in the hip abductor muscles, particularly the gluteus medius (GM), can further result in abnormal stress on the lumbar spine and hip joints, potentially leading to chronic musculoskeletal complications [4]. Krautwurst et al. reported a correlation between abductor strength and trunk obliquity in 375 patients with CP [5]. The authors suggested that an ipsilateral trunk lean towards the affected stance limb—referred to as Duchenne gait—serves as a hip joint unloading mechanism in the presence of weak abductor muscles [5].

In Duchenne gait, subjects move their trunk's center of mass towards the hip joint, reducing the demand on the hip abductors [6]. Gluteus medius as the main hip abductor plays a significant role in stabilizing the pelvis and lower body within the frontal plane during gait [7]. Subsequently, it has a direct influence on altered frontal plane gait, namely Duchenne gait [8]. However, the association between the function of this muscle and the pathological gait seen in CP has not yet been fully established.

Electromyographic (EMG) analysis is an important component in the clinical gait assessment of people with CP, providing a means to evaluate the functionality of their muscles during walking [9]. Davoudi et al. [10] recently presented algorithm clustering EMG data of Rectus femoris as an objective tool to develop treatment decision approaches for patients with CP in crouch gait. Reinbolt et al. [11] also used EMG along with gait kinematics and clinical data to predict the improvement in sagittal plane knee motion in patients with CP following a rectus transfer surgery. On the other hand, Patikas et al. [12] reported minor changes in muscle activity following a single-event multilevel surgery (SEMLS). However, they concluded that the untypical pre-operative EMG patterns might be associated with a compensatory response in some patients. While these studies [12–14] addressed sagittal plane muscles and movement, there is a need for a further analysis on frontal plane muscles such as GM.

Machine learning (ML) is revolutionizing gait analysis by enabling precise identification of gait abnormalities through advanced algorithms such as explainable machine learning, and optimized gait feature extraction methods [15, 16]. Clustering is an unsupervised machine learning approach to determine the main trends in a dataset [17], which can also be applied for the analysis of clinical gait and EMG patterns [10]. Although this analysis basically measures the similarity between the elements and does not require labelled (pre-specified) data, their biomechanical interpretation remains a challenge. Sangeux et al. [14] applied K-means algorithm on the knee and ankle kinematics of patients with CP. They introduced an index for successfully categorizing these patients into different clusters and observed a correlation between these clusters and spasticity in the gastrocnemius-soleus muscles. Using the same technique on the kinematics of lower limb joints in all three planes of motion, Kuntze et al. [17] identified four clinically meaningful clusters. Further, in our recent work [10] we showed a correlation between the results of a hybrid particle swarm optimization (PSO) and K-means clustering on EMG of Rectus femoris and knee flexion-extension angle and moment as well as knee strength and spasticity.

While these studies confirm the applicability of cluster analysis for a better understanding of gait deficits in patients with cerebral palsy, the computational challenges still remain. PSO is an optimization algorithm used to tackle the initialization problem in K-means [18], however, it does not work well when the database is large or complex, for example when increasing the number of muscles [19]. Hierarchical clustering is a method used to group similar objects by constructing a cluster tree, known as a dendrogram. It can be effectively utilized in conjunction with other approaches, such as K-means, to address their respective technical limitations, particularly in the context of large databases. This method starts with considering each data point as separated clusters to eventually link them as a hierarchy. Researchers suggested

applying a combined hierarchical clustering with K-means could increase the performance of the clustering for large and complex datasets [20].

This study investigates the relationship between gluteus medius EMG activity and frontal plane gait abnormalities in the context of CP management, hypothesizing that abnormal GM activation is linked to gait dysfunction and impaired movement efficiency. A key objective is the development of a generalized two-stage hierarchical clustering method to categorize distinct EMG patterns. Additionally, the study aims to examine the connection between EMG of GM and treatment outcomes, particularly in patients displaying Duchenne gait, to determine whether EMG can serve as an indicator to assess the succession the treatment.

## 2. Methods

### 2.1. Participants

Similar to a recent study on Rectus femoris EMG clustering [10], the data analyzed in this retrospective study were part of a larger database established at the local University Clinics in the years 2000–2022 when retrieval was stopped. Only personnel that had regular legal access to the medical records retrieved patient data. They collected data in the time November and December 2022, and anonymized it in the same year December 28th. After this step, individual participants could not be identified anymore. The study was approved by the Ethics Committee of the Medical Faculty of Heidelberg University (Heidelberg, Germany; no: S-243/2022). It waived the requirement for informed patient consent.

According to conventional gait analysis procedures [21], patients had been monitored barefoot in level gait of self-selected speed. Additional retrospective inclusion criteria were the ability to walk without assistive devices, i.e. classified as Gross Motor Functions Classification System (GMFCS) level I and II [22] and the availability of EMG data.

845 patients and 65 typically developed (TD) individuals were recruited for the clustering analysis. The mean and standard deviation (SD) of the participants' characteristics are summarized in Table 1.

### 2.2. Data collection and processing

Raw EMG signals of eight major lower extremity muscles including the Gluteus medius, were bilaterally collected with Bipolar surface adhesive electrodes (Blue Sensor, Ambu Inc., Glen Burnie, MD, USA), using myon 320 (Myon AG, Schwarzenberg, Switzerland). The placement of the electrodes was based on the guidelines provided by SENIAM [23]. The raw signals were band-pass filtered (Butterworth, cutoff frequency of 20–350 Hz); rectified and smoothened (Butterworth low pass, cutoff frequency of 9 Hz); averaged across valid strides; and then time-normalized (within one gait cycle) and amplitude-normalized (to the mean of signal) in Matlab (The MathWorks, Inc. USA) [14]. Twelve cameras (VICON, Oxford Metrics Limited, UK) and three force-plates (Kistler Instruments Co.) were used to capture the trajectories of

**Table 1. Mean (SD) of demographic and descriptive data for participants included in the clustering analysis, as well as TD individuals.**

|  | CP | TD |
|---|---|---|
| Age (years) | 18.5 (9.3) | 20.0 (13.7) |
| Height (cm) | 157.1 (36.3) | 157.1 (19.9) |
| Body mass (kg) | 50.4 (18.6) | 67 (15.3) |
| Sex (Men/Women) | 496/349 | 36/29 |
| CP type (bilateral/unilateral) | 684/161 |  |

the markers and ground reaction force data, respectively. The markers were placed according to the protocol of Kadaba et al. [21]. Applying the plug-in-gait model, the kinematics and kinetics of the lower body joints of the participants were calculated [24]. Hip abduction passive range of motion (RoM) and hip abduction muscles strength in both 0 and 90 degrees (hip flexion position) of the subjects were assessed according to the Medical Research Council (MRC) [25]. In this method the muscle strength is scaled from 5 (full strength) to 0 (no strength). More details about the capturing and analyzing the gait data used in this study is available in our previous works [10, 24, 25].

## 2.3. Cluster analysis

**2.3.1. Feature extraction and standardization.**   After pre-processing EMG data as described above, features were extracted from these time series following previous work [26]. Maximum and minimum values of the time series and their temporal position in the gait cycle, range of motion (= max-min) as well as mean values across the gait cycle were determined. Temporal aspects of these features are further addressed by computing them for the entire stride as well as for relevant sub-phases defined by Perry [27], namely stance, swing, loading response (LR), mid stance (MSt), terminal stance (TSt), pre-swing (PSw), initial swing (ISw), mid swing (MSw), and terminal swing (TSw). The features are available as Supplementary material (S1 Data). These features were then standardized to quantify the deviation of each patient's EMG from TD subjects using Formula (1) [14]. Norm distance (NDi) was defined as the absolute difference between the i$^{th}$ feature of the EMG of the patient p ($F_{pi}$) and the mean value of the same feature in the TD group ($\bar{F}_{ni}$), divided by the corresponding standard deviation within the TD ($SD_{ni}$).

$$NDi = \frac{|F_{pi} - \bar{F}_{ni}|}{SD_{ni}} \tag{1}$$

**2.3.2. K-means-PSO clustering.**   The resulting normalized input matrix has a dimension of 845 (number of patients) × 60 (number of features for 10 gait phases). We applied a hybrid K-means-PSO clustering algorithm to each feature (column) of this matrix. The algorithm's details have been explained in our work on activity of Rectus femoris [10]. In this approach, the output of PSO, as a global optimization search algorithm, served as the initial centroids for the K-means, allowing each datapoint to be assigned to a cluster based on its distance from these centers. Fig 1 visualizes the procedure we used in this stage in a flow chart (Fig 1, stage 1). To evaluate the optimal number of clusters, we utilized the Davies-Bouldin (DB) clustering evaluation criterion in Matlab [28]. The resulted matrix serves for the hierarchical clustering (Fig 1, stage 2).

**2.3.3. Hierarchical clustering.**   Using Principal Component Analysis (PCA) we reduced the dimension of the matrix resulting from the hybrid clustering to ensure that the first principal components (PCs) explained more than 96% of the primary variance [10]. Fig 1 illustrates clustering stage 2, i.e., the flow of the procedure utilized for the hierarchical clustering. We used the Matlab 'Cluster' function to identify the final clusters according to the similarity between the patients. The number of clusters was determined using the 'dendrogram' in Matlab.

## 2.4. Comparison between the clusters

The average activity of the GM in each phase of the gait, along with trunk, pelvic, and hip kinematics and kinetics in the frontal plane, was calculated for the patients identified in each

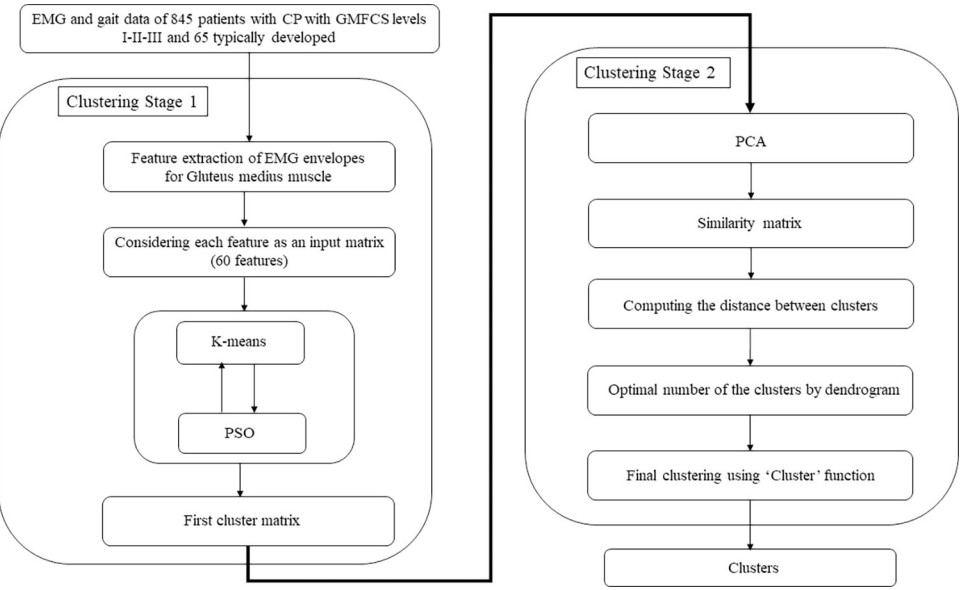

**Fig 1. Flowchart of the two-stage clustering algorithm used in this study.**

cluster. Descriptive statistics (mean and standard deviation) were used to compare the clinical examination data between the conditions.

## 2.5. Identification of the sub_clusters of patients

In this investigation, our primary study cohorts consisted of individuals with CP and TD. Initially, we applied cluster analysis to the patient group, leading to the identification of distinct clusters. Subsequently, focusing on a cluster characterized by pronounced frontal plane features such as excessive lateral trunk lean, we examined changes in EMG and gait between their initial examination (E1) and subsequent examination (E2) in the laboratory setting. As outlined in [29], we assessed the following parameters important for evaluating frontal plane gait in patients with CP: 1) kinematics (range of motion of trunk and pelvic obliquity, maximum hip abduction in mid-stance); 2) kinetics (maximum hip abduction moment in mid-stance), and 3) EMG measures (mean, maximum, and minimum of GM activity) for both E1 and E2 within the selected cluster of patients.

Applying PCA [30] on the 'changes matrix', defined as $\Delta E = parameters@E2 - parameters@E1$, we identified two sub_clusters of patients. Patients with a positive score (PC_scores > 0) were considered as sub_cluster one (SUB_1), and those with a negative score (PC_scores < 0) as sub_cluster two (SUB_2).

These sub-clusters exhibited different responses to the intervention. Therefore, we statistically analyzed the improvement in their frontal plane gait from E1 to E2 using the non-parametric Kruskal-Wallis test to biomechanically characterize the sub-clusters. Additionally, to determine the potential causes of the differing responses, we examined the pre-treatment (E1) features of these patients. The minimum, maximum, and mean EMG activity of GM during 50–87% of the gait cycle, as well as hip abduction strength and range of motion at 0 and 90 degrees hip flexion, were compared between SUB_1 and SUB_2 using a non-parametric test (p-Value = 0.05).

Furthermore, the specifics of the type and number of surgeries performed between E1 and E2 for the different sub_clusters (SUB_1 and 2) were examined to investigate any potential bias arising from the treatment approach on the responses and sub_clustering outcomes.

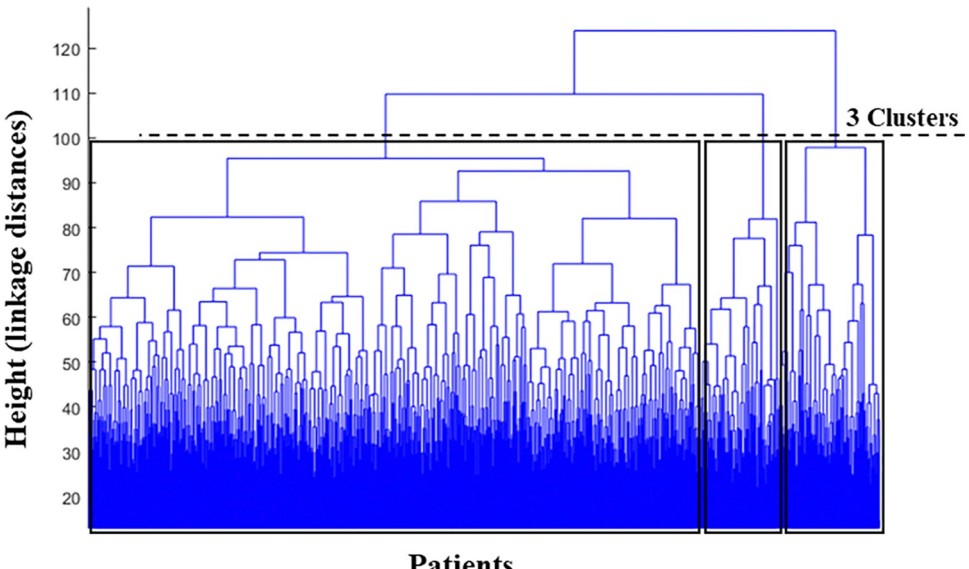

**Fig 2. Detection of three clusters of the patients based on the height (cluster distances) in dendrogram.**

## 3. Results

Using the dendrogram (Fig 2), we identified three main clusters: cluster_1 (84 patients), cluster_2 (654 patients), and cluster_3 (107 patients) (Table 2). Comparing the EMG of the GM and frontal plane kinematics and kinetics between the clusters (Fig 3 and Table 2), we observed that on average, the patients in cluster_1 showed a relatively low activity level of GM during loading response compared to the two other clusters. They (cluster_1) also exhibited an increased lateral trunk lean with a stable pelvic position (Duchenne gait). Patients in this cluster also exhibited a lack of a peak in stance in their hip abduction and hip abductor moment (Fig 3). The largest number of patients (654 out of 845) was classified as cluster_2 in this study with a medium level of GM activity in loading response, while the patients in cluster_3 had the highest activity in this gait phase. Moreover, patients in cluster_3 and cluster_1 had the highest and lowest averaged pre-treatment passive range of hip abduction and of hip abductor strength, respectively (Table 2).

**Table 2. The number of patients in clusters, mean (SD) of averaged EMG patterns of each cluster in different gait phases, and clinical exam data.**

| Clusters | Number of patients | Mean (SD) of EMG envelops in each cluster in different gait subphases | | | | | | | Mean (SD) of hip clinical exam data of patients in each cluster | | | |
|---|---|---|---|---|---|---|---|---|---|---|---|---|
| | | LR | MSt | TSt | PSw | ISw | MSw | TSw | Hip abduction passive RoM (90˚ HF*) | Hip abduction passive RoM (0˚ HF*) | Hip Abductors Strength (90˚ HF*) | Hip Abductors Strength (0˚ HF*) |
| **1** | 84 | 170.1 (40.2) | 126.9 (24.7) | 85.8 (15.8) | 81.3 (22.1) | 72.4 (19.1) | 64.5 (19.4) | 90.8 (28.1) | 32.8 (12.8) | 35.1 (8.6) | 4 (0.9) | 3.6 (0.8) |
| **2** | 654 | 230.2 (59.4) | 145.1 (32.9) | 79.52 (24.1) | 56.1 (27.4) | 46.6 (21.3) | 42.4 (20.4) | 104.5 (35.2) | 35.1 (12.6) | 37.2 (10) | 4.1 (0.8) | 3.8 (0.7) |
| **3** | 107 | 273.6 (61.5) | 135.6 (31.8) | 64.5 (20.8) | 60.5 (33.2) | 49.6 (23.3) | 41.9 (18.4) | 108.1 (41.7) | 41.5 (13.4) | 42.1 (9.9) | 4.5 (0.7) | 4 (0.7) |
| **TD** | 65 | 264.9 (91.3) | 113.3 (36.1) | 71.1 (29.1) | 68.4 (38.1) | 60.1 (28.5) | 57.2 (26.7) | 95.4 (30.4) | 50.4 (10.5) | 49.1 (6.9) | 5 | 5 |

* HF: Hip flexion.

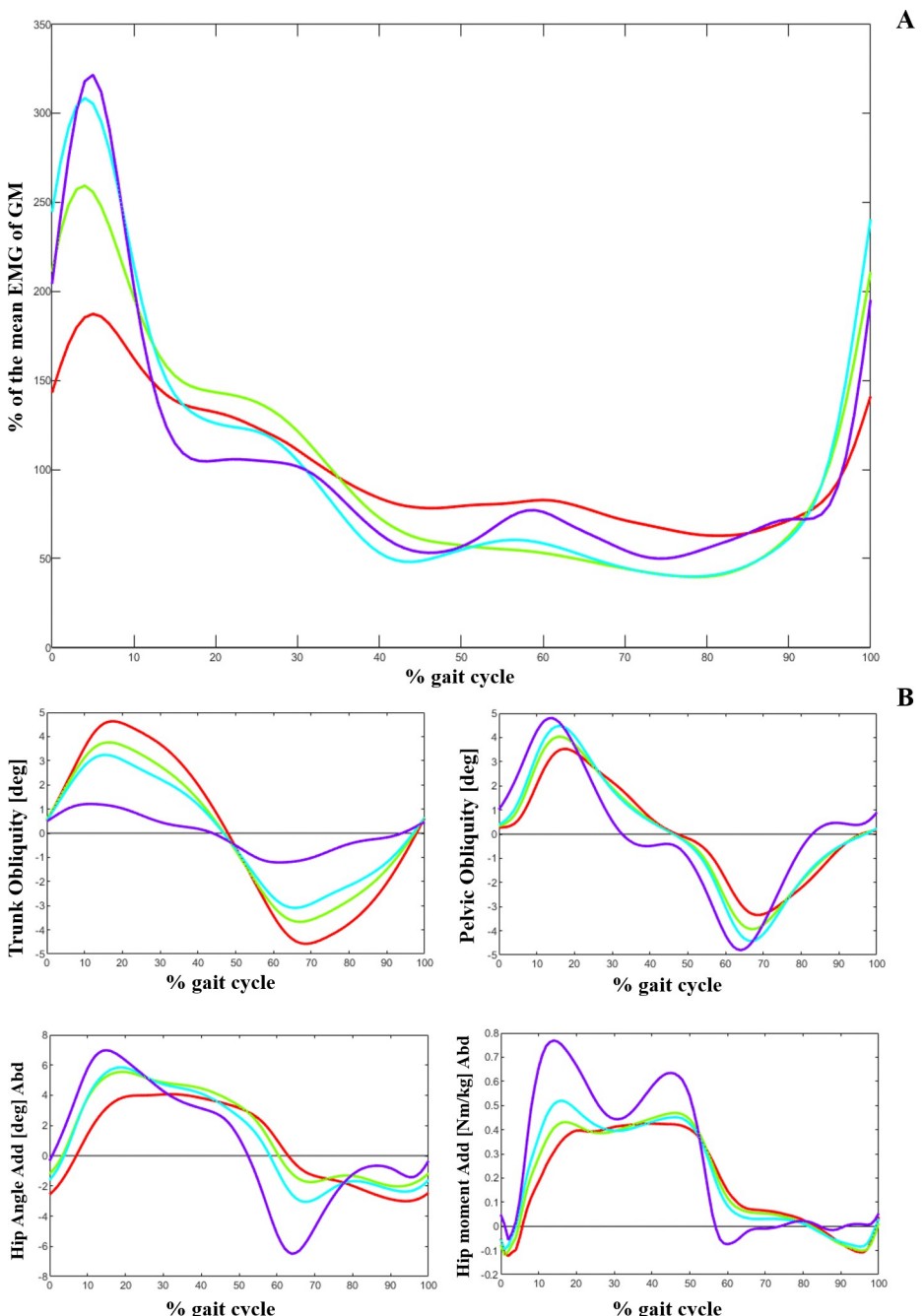

**Fig 3.** A) Average Gluteus medius EMG, B) Trunk, pelvic and hip kinematics and kinetics in frontal plane for 3 different clusters. Red: cluster_1, Green: cluster_2, Blue: cluster_3, Purple: TD.

Some 31 patients (out of 84 (Table 2)) identified as cluster_1 and with pre and post treatment gait and EMG data available were chosen for sub_cluster analysis. The average (SD) PC_scores for SUB_1 (12 patients) and SUB_2 (19 patients) were 127.4 (120.5) and -80.4 (48.7) respectively. Fig 4 shows the sub_clusters identification steps through a flowchart.

Comparing the EMG and frontal plane kinematics patterns between examinations for both sub_clusters (Fig 5 and Table 3), a significant decrease (p = 0.01) in the RoM of trunk obliquity

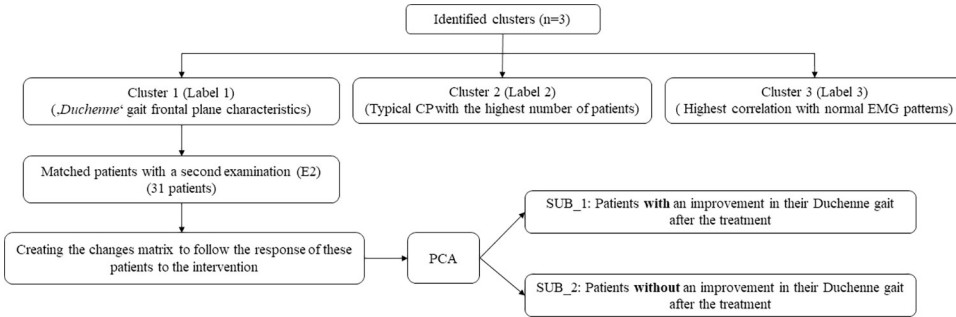

**Fig 4. Identification procedure of the sub_clusters (SUB_1and SUB_2) in patients in cluster_1 with pathological gait.**

was observed in SUB_1. Additionally, an increase in pelvic obliquity and hip abduction angle and moment during midstance was exhibited following the intervention. Therefore, the Duchenne gait abnormality improved in patients who were sub_clusterd as SUB_1 after the intervention. Furthermore, these patients showed a higher pre-treatment activity of GM during 50–87% of their gait (p = 0.05) (Table 4) and a greater RoM of hip abduction at 90 degrees hip flexion (p = 0.05) compared to SUB_2 (Table 4).

Table 5 also shows the details of the main distal and proximal surgeries for the sub_clusters in this study. In total 24 patients (out of 31) underwent a surgery between their first (E1) and second (E2) gait examination. The most frequent distal surgeries (Baumann and Strayer procedures, Tibialis posterior lengthening, and bony foot procedures) and proximal surgeries (femoral derotation, rectus transfer, and hamstring lengthening) were considered for examination between the sub_clusters. In general, the number of the proximal surgeries was relatively higher than distal surgeries for the patients with a significant improved trunk obliquity (SUB_1), while the surgeries around the foot (bony foot procedures) and shank (tibialis posterior lengthening) were more frequent for subjects in SUB_2. Tendo-Achilles lengthening, along with the Baumann and Strayer procedures, was another lengthening technique performed. However, only one patient, identified as part of SUB_2, underwent this specific surgery.

## 4. Discussion

In this study, we introduced a two-stage clustering approach that effectively identified three primary EMG activity patterns of the GM muscle in patients with cerebral palsy. Combined with gait and clinical data, the labels were characterized by a group of patients with a relatively low activity of GM in initial and mid stance and a pathological (Duchenne) gait pattern (cluster_1); an increased activity in mid-stance with the highest number of identified patients (cluster_2); and the highest EMG activity during loading response with the strongest hip abductors (cluster_3). The results were consistent with our latest work on the Rectus femoris EMG clustering, in which we categorized patients into pathological (crouch) gait, typical CP, and correlated-to-TD patterns [10]. To achieve this, we applied a pre-clustering standardization to transform the features into their deviation from a healthy population. This approach allowed us to obtain meaningful and reliable clustering results [10, 12].

The patients in cluster_1 who showed weakest hip abductors and decreased hip abduction RoM, demonstrated a more pronounced trunk lean with a decreased pelvic RoM (Fig 4, Table 2). This finding aligns with the results of Krautwurst et al. [5] who observed a correlation between decreased abductor muscle strength and trunk and pelvic kinematics. Furthermore,

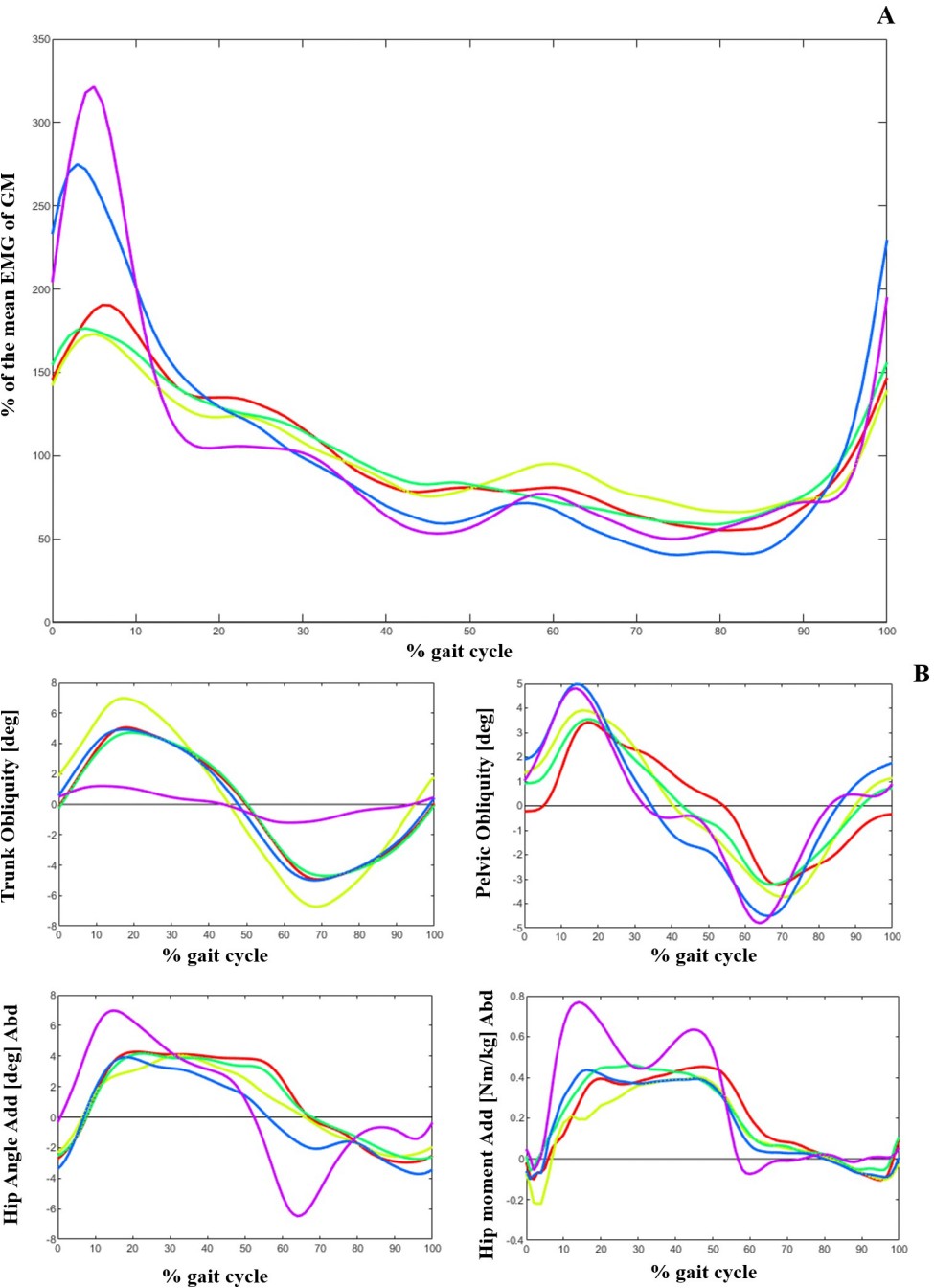

**Fig 5.** A) Average Gluteus medius EMG, B) Trunk, pelvic and hip kinematics and kinetics in frontal plane for patients of SUB_1and SUB_2 in their E1 and E2. Yellow: SUB_1-E1, Red: SUB_2-E1, Blue: SUB_1-E2, Green: SUB_2-E2, Purple: TD.

they observed an increased trunk lateral obliquity, serving as a compensatory mechanism to stabilize the pelvis in the frontal plane and keep the gait stable in patients with CP. The hip unloading observed in patients identified as cluster_1, in the presence of hip abductor weakness, is a characteristic finding of Duchenne gait [31], which shifts the body's center of mass laterally and significantly reduces the hip abduction moments. However, a substantial increase

**Table 3. Statistical comparison of the pre- and post-treatment frontal plane gait parameters of the patients sub-clustered as SUB_1 and SUB_2.**

| | RoM Trunk Obliquity [deg] | | RoM Pelvic Obliquity [deg] | | Max Hip Abd/Add Angle in MSt [deg] | | Max Hip Abd/Add Moment in MSt [Nm/kg] | |
|---|---|---|---|---|---|---|---|---|
| | SUB_1 | SUB_2 | SUB_1 | SUB_2 | SUB_1 | SUB_2 | SUB_1 | SUB_2 |
| E1 | 15.4 (7.4) | 11.5 (4.4) | 10.1 (3.5) | 9.5 (4.6) | 5.8 (3.9) | 6 (3.9) | 0.46 (0.25) | 0.52 (0.18) |
| E2 | 10.5 (5.4) | 10.8 (5.8) | 11.9 (4.7) | 8.9 (4.7) | 6.5 (3.5) | 5.3 (4.3) | 0.52 (0.22) | 0.55 (0.14) |
| p-value between E1 and E2 within each group | **0.01*** | 0.29 | 0.24 | 0.39 | 0.76 | 0.51 | 0.4 | 0.36 |
| TD | 2.9 (1.4) | | 9.8 (3) | | 7.2 (2.6) | | 0.79 (0.16) | |

* p≤ 0.05.

in the requirement for effort and work by the trunk muscles is the price for this compensation [31]. EMG studies show the greatest burst of GM occurring during loading response and initial stance [7]. In our study also, the main distinction between the cluster patterns was visually observed in this phase of the gait. However, notably, the higher averaged EMG activity in swing phase of patients in cluster_1 might be attributed to the increased tightness of GM in these individuals. The anterior part of the GM assists in hip flexion and the activity of this muscle during the swing phase could potentially contribute to lengthening the step and promoting forward movement.

Investigating the relationship between EMG activity and altered trunk obliquity in sub_-clustered patients (SUB_1 and SUB_2) (Fig 5, Table 3), we observed that an increased activity of GM during 50–87% of gait cycle, as well as a greater RoM of hip abduction, were the significant indicators of treatment outcomes for the Duchenne condition. Considering the discussion presented by Heyrman et al. [32], an altered trunk movement observed in CP, such as Duchenne gait, is not exclusively a compensatory response to lower limb impairments. For certain patients, it could reflect a deficit in trunk motor control. We suggest that the patients identified as SUB_1 and SUB_2 can potentially represent the compensatory and motor control related Duchenne gait, respectively. Therefore, after treatment and fixing the biomechanical problem, subsequently the frontal plane movement of trunk in SUB_1 significantly improved. In addition, the compensatory prolonged activation of GM during swing (50–87% of gait) was reduced in these patients (SUB_1) after the intervention. Employing PCA, Rethwilm et al. [30] also concluded that Duchenne gait in CP could arise from motor disfunction rather than being only a compensatory strategy which supports our suggestion. However, while assessment of the motor control of the patients was not an aim of this study, we recommend it for future studies to outline the influence of the trunk motor deficit on Duchenne gait.

**Table 4. Statistical comparison of the pretreatment EMG levels and physical examination data between the patients sub-clustered as SUB_1 and SUB_2.**

| Group | EMG | | | Physical Examination | | | |
|---|---|---|---|---|---|---|---|
| | Mean | Min | Max | Hip abduction passive RoM (90˚ HF*) | Hip abduction passive RoM (0˚ HF*) | Hip Abductors Strength (90˚ HF*) | Hip Abductors Strength (0˚ HF*) |
| | 50–87% | 50–87% | 50–87% | | | | |
| SUB_1-E1 | 79.5 (19.2) | 50.7 (15.1) | 111.7 (31.9) | 37.2 (12.5) | 35.8 (7.9) | 3.9 (0.5) | 3.5 (0.5) |
| SUB_2-E1 | 68.6 (8.7) | 45.5 (10.8) | 99.1 (21.9) | 29.6 (10.3) | 32.7 (7.1) | 4 (1) | 3.4 (0.6) |
| P-value | **0.05*** | 0.39 | 0.25 | **0.05*** | 0.35 | 0.8 | 0.97 |

* p≤ 0.05.
** HF: Hip flexion.

**Table 5. Comparison between the type and number of surgeries between sub_clusters.**

| Responders | Total number of patients | No surgery | Distal surgeries | | | | Proximal surgeries | | | total number of distal surgeries | total number of proximal surgeries |
|---|---|---|---|---|---|---|---|---|---|---|---|
| | | | Baumann procedure | Strayer procedure | bony foot procedures | Tibialis posterior lengthening | femoral derotation | rectus transfer | hamstring lengthening | | |
| SUB_1 | 12 | 3 | 4 | 2 | 3 | 1 | 6 | 3 | 3 | 10 | 12 |
| SUB_2 | 19 | 4 | 6 | 3 | 9 | 5 | 14 | 5 | 3 | 23 | 22 |
| Total | 31 | 7 | 10 | 5 | 12 | 6 | 20 | 8 | 6 | 33 | 34 |

It is important to note that therapy for patients with CP typically begins in early childhood and continues throughout their lifetime. Additionally, patients often undergo multiple corrective procedures addressing soft tissue and bone deformities in a single orthopedic intervention, referred to as single event multilevel surgery. These surgeries are frequently performed by different surgeons at various stages, which adds variability to the treatment process. Consequently, when working with large databases, it becomes clinically challenging to isolate the effects of factors such as the type and number of treatments, the expertise of the surgeon or clinician, and other contextual parameters. In the current study, the facts that Duchenne gait primarily involves the trunk and that patients in SUB_1 underwent more proximal surgeries than distal surgeries (Table 5) may introduce a potential bias in the sub-clustering approach we employed in this study. This study primarily focused on the applicability of the EMG of the gluteus medius as a clinical measure to enhance decision-making for patients with CP. However, future studies could investigate the impact of different treatment approaches, including the number and type of surgeries, on the EMG and Duchenne gait of patients with CP.

To the best of our knowledge, our study is the first to investigate the relationship between the activity of the GM muscle as measured by surface EMG and the kinematics of the trunk and legs in the frontal plane of patients with CP, using a clustering approach. Our findings suggest that the EMG of the gluteus medius may be associated with improvements in Duchenne gait following treatment. We recommend that clinicians evaluate GM activity during the stance-to-swing transition phase and measure passive hip abduction RoM at 90˚ hip flexion prior to surgery for patients with Duchenne gait. An EMG activity level below 70% of the mean, combined with a restricted RoM (<30˚), may indicate that surgery alone is unlikely to result in improved trunk movement. Such patients may benefit from additional interventions, such as trunk motor control training, before undergoing surgical treatment.

Specifically, we conclude that a passive hip abduction RoM <30˚ likely indicates a structural issue that may require orthopedic treatment independently of other factors. Conversely, if RoM >30˚ and EMG activity exceeds 70% during the 50–87% phase of the gait cycle, there is a strong likelihood that the patient will experience reduced active trunk lateral movement after treatment. This outcome suggests effective motor control of the hip abductors during the stance-to-swing transition. However, if RoM >30˚ and EMG activity remains below 70%, there is no structural problem but motor control/spasticity is rather poor. In such cases, the likelihood of improvement in Duchenne gait after treatment is limited.

These recommendations underscore the potential of EMG as a predictor of treatment outcomes in the context of gait. However, this clinical implication is derived from data collected and analyzed within our center and is limited to similar clinical settings. Future studies are necessary to validate these findings and further explore their applicability by investigating other patients with cerebral palsy and Duchenne gait in diverse clinical environments.

In this study, we employed the Davies-Bouldin (DB) criterion to determine the optimal number of clusters. Alternative methods, such as the elbow diagram [33], contour coefficient

value [34], and silhouette score [35], are also available for this purpose and could be explored in future studies. Given that our results were biomechanically meaningful, we advocate for the DB criterion as a robust approach. Furthermore, although we had access to a relatively large database of EMG patterns, our analysis focused exclusively on a single muscle. Future research could extend this work by applying the two-stage clustering method proposed here to a group of synergistic muscles. Finally, while the number of patients included in this study is already substantial, future studies could benefit from longer follow-up data or multicenter designs with larger sample sizes. Furthermore, while this study focused on EMG envelopes, it is recommended that future research includes the calculation of EMG and motor control indices [36] and compares these metrics across patients. Such indices can summarize EMG signal information into clinically interpretable formats, enhancing their applicability in treatment management.

## Supporting information

**S1 Data.**
(XLSX)

## Author Contributions

**Conceptualization:** Firooz Salami, Katharina S. Gather, Sebastian I. Wolf.

**Data curation:** Mehrdad Davoudi, Firooz Salami, Robert Reisig.

**Formal analysis:** Firooz Salami.

**Funding acquisition:** Sebastian I. Wolf.

**Investigation:** Mehrdad Davoudi, Sebastian I. Wolf.

**Methodology:** Mehrdad Davoudi, Firooz Salami, Sebastian I. Wolf.

**Project administration:** Sebastian I. Wolf.

**Resources:** Katharina S. Gather, Sebastian I. Wolf.

**Software:** Mehrdad Davoudi, Firooz Salami, Robert Reisig.

**Supervision:** Firooz Salami, Katharina S. Gather, Sebastian I. Wolf.

**Validation:** Mehrdad Davoudi.

**Visualization:** Mehrdad Davoudi.

**Writing – original draft:** Mehrdad Davoudi.

**Writing – review & editing:** Firooz Salami, Robert Reisig, Katharina S. Gather, Sebastian I. Wolf.

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
