## [Decision Letter · Decision Letter 0]

13 Nov 2024

PONE-D-24-34676Gluteus Medius Muscle Activation Patterns during Gait with Cerebral Palsy (CP): A hierarchical clustering analysisPLOS ONE

Dear Dr. Wolf,

Thank you for submitting your manuscript to PLOS ONE. After careful consideration, we feel that it has merit but does not fully meet PLOS ONE’s publication criteria as it currently stands. Therefore, we invite you to submit a revised version of the manuscript that addresses the points raised during the review process.

We look forward to receiving your revised manuscript.

Kind regards,

Yaodong Gu

Academic Editor

PLOS ONE

**Journal Requirements:**

German Research Foundation (DFG)

no: WO 1624/ 8-1

**Additional Editor Comments:**

Most part of the discussion part shall be re-written as reviewer's suggestion.

Reviewers' comments:

Reviewer's Responses to Questions

**Comments to the Author**

1. Is the manuscript technically sound, and do the data support the conclusions?

Reviewer #1: Partly

Reviewer #2: Yes

2. Has the statistical analysis been performed appropriately and rigorously? 

Reviewer #1: N/A

Reviewer #2: Yes

3. Have the authors made all data underlying the findings in their manuscript fully available?

Reviewer #1: Yes

Reviewer #2: Yes

4. Is the manuscript presented in an intelligible fashion and written in standard English?

Reviewer #1: Yes

Reviewer #2: Yes

5. Review Comments to the Author

**Reviewer #1:** The manuscript, titled " Gluteus Medius Muscle Activation Patterns during Gait with Cerebral Palsy (CP): A hierarchical clustering analysis " aims to investigate in depth the relationship between hip abductors (especially gluteus medius) and Duchenne gait in patients with cerebral palsy (CP) employing electromyographic (EMG) cluster analysis. Gait data from 845 CP patients and 65 normally developing individuals were analyzed and different EMG activity patterns were found to be associated with Duchenne gait characteristics.

However, in the introduction section, it is recommended that a discussion of the clinical importance of the impact of CP and hip abductor dysfunction on children's daily living and motor abilities be added. Furthermore, the Methods section should detail the participant's gait assessment criteria and how the six characteristics relate to the research question. Also, describe how coaching bias was minimized and other clustering assessment criteria were considered. Moreover, the discussion needs to be supplemented with an analysis of the study limitations and recommendations for clarifying future research directions, such as longer follow-up data and multicenter large-sample studies. Specific comments are shown below:

Introduction

1. Lines 49-54: When referring to the effects of CP and hip abductor dysfunction, a discussion of the clinical importance of the problem could be added, including the impact on the child's daily life and motor abilities. This would help the reader understand the need for the study.

2. Lines 72-83: Regarding the application of machine learning methods in gait analysis, it is necessary to provide a detailed overview of the latest research directions (including explainable machine learning, optimized gait feature extraction algorithms, etc.), further drawing out the advantages of cluster analysis. To provide more effective evidence, the authors may consider referring to the following relevant studies: A new method proposed for realizing human gait pattern recognition: Inspirations for the application of sports and clinical gait analysis (https://doi.org/10.1016/j.gaitpost.2023.10.019); Explaining the differences of gait patterns between high and low‑mileage runners with machine learning (https://doi.org/10.1038/s41598-022-07054-1)

3. Lines 95-98: In the last paragraph, the main research questions and hypotheses of the study are clearly and unambiguously listed so that the reader can understand the direction and objectives of the study at a glance.

Methods

1. Lines 108: You mentioned “ability to walk barefoot without any assistive device”, could you elaborate on how participants were tested in the gait assessment? For example, were standardized gait testing procedures performed?

2. In describing the six main features extracted, it is possible to further explain how these features specifically relate to the research question and why they were chosen for cluster analysis.

3. Describe how the presence of multiple certified strength and conditioning specialists minimizes coaching bias. This is important for ensuring that training is administered uniformly across groups.

4. In determining the optimal number of clusters, the Davies-Bouldin criterion was used; are there other assessment criteria (e.g., contour coefficients) for comparison to ensure the reliability of the selected clusters?

Discussion

Your discussion section does not mention the limitations of the current study. Is it possible to add a discussion of sample size, sample characteristics, experimental design, etc.? Also, it is recommended to clarify specific directions for future studies, such as considering longer follow-up data or multicenter studies with larger samples.

**Reviewer #2: **My overall assessment of the reviewed work is very good, I recommend submitting the manuscript for publication without the need for major changes.

The research topic is well-grounded in the literature review and current needs of human movement analysis. The use of an existing database and its more comprehensive analysis, using a new patient classification method proposed by the authors, should be considered a plus. The data was created using top-class measurement tools and methods (Vicon system, Kistler platforms), which guarantees an appropriate level of data quality. The presented research and data analysis protocol is described in sufficient detail, which allows it to be reproduced by other research teams who would like to use the proposed technique in their projects.

The presented results correlate well with the stated goals of the work and allow for proper conclusions. I have minor comments on this part of the work, which I will present in detail at the end of the review. It also seems necessary to indicate more clearly the limitations of the work that may affect the obtained results.

I would also like to point out that the abstract of the work with the same/similar title is publicly available as conference materials from 2024 and I leave it to the Publisher's decision whether this does not violate the declaration of submitting work not published in any other journal.

Minor detailed comments:

Table with participants characteristics – what values are in the brackets? (table 1)

Table 2 - why come from number of TD subjects equal 117? In table 1, also in paper text is 65 typical development subjects.

10 gait phases – including a full cycle. It my opinion would be better to describe it as “... for 10 time intervals including a full gait cycle and its 9 phases: LR, etc….”. Writing about a full gait cycle as a phase of it is a bit misleading. But it's only me feeling.

Line 183-185: "The 50-87% of the gait cycle (pre-, initial, and mid-swing phases) was visually chosen by comparing the EMG patterns." – I don't fully understand this sentence. Does this mean that by visually looking at the emg pattern it was determined when the preswing, initial, and mid-swing occurred, without using normalized time values? Why? What was the rules for 'visually determined intervals?

Line 225:"Furthermore, these patients showed a higher pre-treatment activity of GM during 50-87% of their gait (p=0.05) (Table 3) and a greater RoM of hip abduction at 90 degrees hip flexion (p=0.05) compared to SUB_2 (Table 4)."

Please check what the tables talk about, table 3 is not about emg.

Line 283: "Therefore, after treatment and fixing the biomechanical problem, subsequently the frontal plane movement of pelvic and trunk in SUB_1 improve."

Please note that only a significant difference was confirmed for the trunk range of motion, so information about the pelvis cannot be added here, it is unconfirmed.

Line 285: "In addition, the compensatory prolonged activation of GM during swing (50-87% of gait) was diminished in these patients (SUB_1) after the intervention.”

I don't think this statement is based on facts. As is known, the EMG signal is characterized by a certain variability, about which I have no information in the referenced Figure 5. It would be necessary to present a graph separately for Sub1 and Sub2 with the muscle activity profile (its average line but necessarily with the standard deviation range) during the observation of E1 and E2. It may turn out that looking at the average signal and its variability, both time points of observation of the EMG profile do not differentiate (and this should be assumed in the visual assessment, if the average +/- STD is superimposed on another compared course also expressed as the average +/- STD.) The authors also did not indicate any statistical test confirming their conclusion for the indicated range of the gait cycle (50-87%) comparing muscle activity before and after therapy/surgical intervention.

Line 304:"An activity level above 70% of mean EMG, combined with a limited range of motion of less than 30 degrees, may suggest that surgery alone will not automatically improve trunk movement, and the patient may require additional treatment such as trunk motor control training, prior to surgery."

Does this mean that if the patient who has less than 70% and a limited range of motion below 30 degrees, that is a good sign for them? Or maybe a patient can have a low EMG level but a good range and that is already ok? It is necessary to clearly state here the predictor of probably insufficient improvement and what values may indicate a probable better final result of the entire therapy. Your take home message should be precise.

6. PLOS authors have the option to publish the peer review history of their article (what does this mean?). If published, this will include your full peer review and any attached files.

Reviewer #1: No

Reviewer #2: **Yes: **Grzegorz Sobota

---

## [Author Response · Author response to Decision Letter 0]

5 Dec 2024

PONE-D-24-34676

Response to reviewers

Editor Dr. Yaodong Gu: Most part of the discussion part shall be re-written as reviewer's suggestion.

Authors: We sincerely appreciate the editor's effort in reviewing our manuscript. response to the comments provided, we have thoroughly addressed the concerns throughout the manuscript, particularly by discussing the study's limitations and offering recommendations for future research directions.

Reviewer #1: The manuscript, titled " Gluteus Medius Muscle Activation Patterns during Gait with Cerebral Palsy (CP): A hierarchical clustering analysis " aims to investigate in depth the relationship between hip abductors (especially gluteus medius) and Duchenne gait in patients with cerebral palsy (CP) employing electromyographic (EMG) cluster analysis. Gait data from 845 CP patients and 65 normally developing individuals were analyzed and different EMG activity patterns were found to be associated with Duchenne gait characteristics. However, in the introduction section, it is recommended that a discussion of the clinical importance of the impact of CP and hip abductor dysfunction on children's daily living and motor abilities be added. Furthermore, the Methods section should detail the participant's gait assessment criteria and how the six characteristics relate to the research question. Also, describe how coaching bias was minimized and other clustering assessment criteria were considered. Moreover, the discussion needs to be supplemented with an analysis of the study limitations and recommendations for clarifying future research directions, such as longer follow-up data and multicenter large-sample studies. 

Authors: Thank you very much for reviewing our study. We have carefully revised our manuscript and addressed your suggestions by making improvements to the introduction, methods, and discussion sections.

Introduction

Reviewer #1: 1. Lines 49-54: When referring to the effects of CP and hip abductor dysfunction, a discussion of the clinical importance of the problem could be added, including the impact on the child's daily life and motor abilities. This would help the reader understand the need for the study.

Authors: Thank you very much for the suggestion. We have revised the introduction by adding the following paragraph and relevant references:

“Hip abductor muscle dysfunction …. It leads to increased energy demands during walking, which can reduce endurance and limit participation in daily activities [2]. Furthermore, it may induce functional restrictions, hindering age-appropriate tasks and basic activities of daily living such as walking, dressing, and playing [3]. Weakness or spasticity in the hip abductor muscles, particularly the gluteus medius (GM), can further result in abnormal stress on the lumbar spine and hip joints, potentially leading to chronic musculoskeletal complications [4].”

Reviewer #1: 2. Lines 72-83: Regarding the application of machine learning methods in gait analysis, it is necessary to provide a detailed overview of the latest research directions (including explainable machine learning, optimized gait feature extraction algorithms, etc.), further drawing out the advantages of cluster analysis. To provide more effective evidence, the authors may consider referring to the following relevant studies: A new method proposed for realizing human gait pattern recognition: Inspirations for the application of sports and clinical gait analysis (https://doi.org/10.1016/j.gaitpost.2023.10.019); Explaining the differences of gait patterns between high and low mileage runners with machine learning (https://doi.org/10.1038/s41598-022-07054-1).

Authors: Thank you. We have modified our introduction with the suggested references:

“Machine learning is revolutionizing gait analysis by enabling precise identification of gait abnormalities through advanced algorithms such as explainable machine learning, and optimized gait feature extraction methods [15, 16]. Clustering is an unsupervised machine learning approach to determine the main trends in a dataset [17], which can also be applied for the analysis of clinical gait and EMG patterns [10]. Although this analysis basically measures the similarity between the elements and does not require labelled (pre-specified) data, their biomechanical interpretation remains a challenge.“

Reviewer #1: 3. Lines 95-98: In the last paragraph, the main research questions and hypotheses of the study are clearly and unambiguously listed so that the reader can understand the direction and objectives of the study at a glance.

Authors: We have revised the last paragraph of the introduction as follows:

“This study investigates the relationship between gluteus medius EMG activity and frontal plane gait abnormalities in the context of CP management, hypothesizing that abnormal GM activation is linked to gait dysfunction and impaired movement efficiency. A key objective is the development of a generalized two-stage hierarchical clustering method to categorize distinct EMG patterns. Additionally, the study aims to examine the connection between EMG of GM and treatment outcomes, particularly in patients displaying Duchenne gait, to determine whether EMG can serve as an indicator to assess the succession the treatment.”

Methods

Reviewer #1: 1. Lines 108: You mentioned “ability to walk barefoot without any assistive device”, could you elaborate on how participants were tested in the gait assessment? For example, were standardized gait testing procedures performed?

Authors: We clarified this as following:

“According to conventional gait analysis procedures [21], patients had been monitored barefoot in level gait of self-selected speed. Additional retrospective inclusion criteria were the ability to walk without assistive devices, i.e. classified as Gross Motor Functions Classification System (GMFCS) level I and II [22] and the availability of EMG data.”

Reviewer #1: 2. In describing the six main features extracted, it is possible to further explain how these features specifically relate to the research question and why they were chosen for cluster analysis.

Authors: This wording was slightly misleading. In fact, it is rather a large number of features for obtaining dense information of EMG data. We rephrased as following:

“After pre-processing EMG data as described above, features were extracted from these time series following previous work [27]. Maximum and minimum values of the time series and their temporal position in the gait cycle, range of motion (=max-min) as well as mean values across the gait cycle were determined. Temporal aspects of these features are further addressed by computing them for the entire stride as well as for relevant sub-phases defined by Perry [28], namely stance, swing, loading response (LR), mid stance (MSt), terminal stance (TSt), pre-swing (PSw), initial swing (ISw), mid swing (MSw), and terminal swing (TSw).”

Reviewer #1: 3. Describe how the presence of multiple certified strength and conditioning specialists minimizes coaching bias. This is important for ensuring that training is administered uniformly across groups.

Authors: The authors believe this comment is not applicable to the current study. Terms such as "coaching" and "training" are not relevant, as we developed an unsupervised clustering algorithm. Therefore, there was no division into test and training sets in our data analysis.

Furthermore, from a clinical perspective, it is important to note that therapy for patients with CP typically begins in early childhood and continues throughout their lifetime. These patients often undergo multiple corrective procedures addressing soft tissue and bone deformities within a single orthopedic intervention, known as single event multilevel surgery (SEMLS). These procedures are frequently performed by different surgeons at various stages, introducing variability into the treatment process.

When working with large databases, it becomes clinically challenging to isolate the effects of factors such as the type and number of treatments, the expertise of the surgeon or clinician, and other contextual parameters. This study primarily aimed to demonstrate how EMG can serve as a critical source of information for clinical decision-making, while acknowledging that these other parameters remain significant.

We have added the following yellow-highlighted paragraph into the discussion:

“It is important to note that therapy for patients with CP typically begins in early childhood and continues throughout their lifetime. Additionally, patients often undergo multiple corrective procedures addressing soft tissue and bone deformities in a single orthopedic intervention, referred to as single event multilevel surgery. These surgeries are frequently performed by different surgeons at various stages, which adds variability to the treatment process. Consequently, when working with large databases, it becomes clinically challenging to isolate the effects of factors such as the type and number of treatments, the expertise of the surgeon or clinician, and other contextual parameters.” In the current study, the facts that Duchenne gait primarily involves the trunk and that patients in SUB_1 underwent more proximal surgeries than distal surgeries (Table 5) may introduce a potential bias in the sub-clustering approach we employed in this study. This study primarily focused on the applicability of the EMG of the gluteus medius as a clinical measure to enhance decision-making for patients with CP. However, future studies could investigate the impact of different treatment approaches, including the number and type of surgeries, on the EMG and Duchenne gait of patients with CP.

Reviewer #1: 4. In determining the optimal number of clusters, the Davies-Bouldin criterion was used; are there other assessment criteria (e.g., contour coefficients) for comparison to ensure the reliability of the selected clusters?

Authors: Thank you for the suggestion. However, given that our results were biomechanically meaningful when using the DNB index, we see no need to calculate and compare different approaches for detecting the number of clusters. To address this point, we have added the following paragraph to the discussion section:

“In this study, we employed the Davies-Bouldin (DB) criterion to determine the optimal number of clusters. Alternative methods, such as the elbow diagram [35], contour coefficient value [36], and silhouette score [37], are also available for this purpose and could be explored in future studies. Given that our results were biomechanically meaningful, we advocate for the DB criterion as a robust approach.”

Discussion

Reviewer #1: 1. Your discussion section does not mention the limitations of the current study. Is it possible to add a discussion of sample size, sample characteristics, experimental design, etc.? Also, it is recommended to clarify specific directions for future studies, such as considering longer follow-up data or multicenter studies with larger samples.

Authors: Thank you for the suggestion. We have added the following paragraph at the end of discussion to address this concern:

“In this study, we employed the Davies-Bouldin (DB) criterion to determine the optimal number of clusters. Alternative methods, such as the elbow diagram [35], contour coefficient value [36], and silhouette score [37], are also available for this purpose and could be explored in future studies. Given that our results were biomechanically meaningful, we advocate for the DB criterion as a robust approach. Furthermore, although we had access to a relatively large database of EMG patterns, our analysis focused exclusively on a single muscle. Future research could extend this work by applying the two-stage clustering method proposed here to a group of synergistic muscles. Finally, while the number of patients included in this study is already substantial, future studies could benefit from longer follow-up data or multicenter designs with larger sample sizes. Furthermore, while this study focused on EMG envelopes, it is recommended that future research includes the calculation of EMG and motor control indices [38] and compares these metrics across patients. Such indices can summarize EMG signal information into clinically interpretable formats, enhancing their applicability in treatment management.”

Reviewer #2: My overall assessment of the reviewed work is very good, I recommend submitting the manuscript for publication without the need for major changes.

The research topic is well-grounded in the literature review and current needs of human movement analysis. The use of an existing database and its more comprehensive analysis, using a new patient classification method proposed by the authors, should be considered a plus. The data was created using top-class measurement tools and methods (Vicon system, Kistler platforms), which guarantees an appropriate level of data quality. The presented research and data analysis protocol is described in sufficient detail, which allows it to be reproduced by other research teams who would like to use the proposed technique in their projects.

The presented results correlate well with the stated goals of the work and allow for proper conclusions. I have minor comments on this part of the work, which I will present in detail at the end of the review. It also seems necessary to indicate more clearly the limitations of the work that may affect the obtained results.

I would also like to point out that the abstract of the work with the same/similar title is publicly available as conference materials from 2024 and I leave it to the Publisher's decision whether this does not violate the declaration of submitting work not published in any other journal.

Authors: Thank you for this recommendation. We present some of this material as an abstract to the conference ESMAC (www.esmac.org) however, this is not to be regarded as a full original article. 

Reviewer #2: Minor detailed comments:

1. Table with participants characteristics – what values are in the brackets? (table 1)

Authors: Thank you very much. The values were the standard deviations. We revised table 1.

Reviewer #2: 2. Table 2 - why come from number of TD subjects equal 117? In table 1, also in paper text is 65 typical development subjects.

Authors: Thank you for spotting this error. We corrected the number in Table 2 to 65. 

Reviewer #2: 3. 10 gait phases – including a full cycle. It my opinion would be better to describe it as “... for 10 time intervals including a full gait cycle and its 9 phases: LR, etc….”. Writing about a full gait cycle as a phase of it is a bit misleading. But it's only me feeling.

Authors: We revised the paragraph according to the suggestion:

“After pre-processing EMG data as described above, features were extracted from these time series following previous work [27]. Maximum and minimum values of the time series and their temporal position in the gait cycle, range of motion (=max-min) as well as mean values across the gait cycle were determined. Temporal aspects of these features are further addressed by computing them for the entire stride as well as for relevant sub-phases defined by Perry [28], namely stance, swing, loading response (LR), mid stance (MSt), terminal stance (TSt), pre-swing (PSw), initial swing (ISw), mid swing (MSw), and terminal swing (TSw).”

Reviewer #2: 4. Line 183-185: "The 50-87% of the gait cycle (pre-, initial, and mid-swing phases) was visually chosen by comparing the EMG patterns." – I don't fully understand this sentence. Does this mean that by visually looking at the emg pattern it was determined when the preswing, initial, and mid-swing occurred, without using normalized time values? Why? What was the rules for 'visually determined intervals?

Authors: Thank you for pointing this out. The wording was indeed slightly misleading. The gait events were determined based on foot events, following our routine gait analysis protocol in the lab. By reviewing the EMG graphs of the patients, we specifically chose the stance-swing transition phase (50-87% of the gait cycle) for further statistical comparisons between the EMG of the sub-clusters. To avoid any misunderstanding, we have removed this sentence from the manuscript

---

## [Decision Letter · Decision Letter 1]

20 Dec 2024

Gluteus Medius Muscle Activation Patterns during Gait with Cerebral Palsy (CP): A hierarchical clustering analysis

PONE-D-24-34676R1

Dear Dr. Wolf,

We’re pleased to inform you that your manuscript has been judged scientifically suitable for publication and will be formally accepted for publication once it meets all outstanding technical requirements.

Kind regards,

Yaodong Gu

Academic Editor

PLOS ONE

Additional Editor Comments (optional):

Reviewers' comments:

Reviewer's Responses to Questions

**Comments to the Author**

1. If the authors have adequately addressed your comments raised in a previous round of review and you feel that this manuscript is now acceptable for publication, you may indicate that here to bypass the “Comments to the Author” section, enter your conflict of interest statement in the “Confidential to Editor” section, and submit your "Accept" recommendation.

Reviewer #1: (No Response)

Reviewer #2: All comments have been addressed

2. Is the manuscript technically sound, and do the data support the conclusions?

Reviewer #1: (No Response)

Reviewer #2: (No Response)

3. Has the statistical analysis been performed appropriately and rigorously? 

Reviewer #1: (No Response)

Reviewer #2: (No Response)

4. Have the authors made all data underlying the findings in their manuscript fully available?

Reviewer #1: (No Response)

Reviewer #2: (No Response)

5. Is the manuscript presented in an intelligible fashion and written in standard English?

Reviewer #1: (No Response)

Reviewer #2: (No Response)

6. Review Comments to the Author

Reviewer #1: All comments have been addressed.

Reviewer #2: (No Response)

7. PLOS authors have the option to publish the peer review history of their article (what does this mean?). If published, this will include your full peer review and any attached files.

Reviewer #1: No

Reviewer #2: **Yes: **Grzegorz Sobota

---

## [Editor Report · Acceptance letter]

30 Dec 2024

PONE-D-24-34676R1 

PLOS ONE

Dear Dr. Wolf, 

I'm pleased to inform you that your manuscript has been deemed suitable for publication in PLOS ONE. Congratulations! Your manuscript is now being handed over to our production team.

Kind regards, 

on behalf of

Professor Yaodong Gu 

Academic Editor

PLOS ONE